# Bioinformatics and Experimental Analyses Reveal Immune-Related LncRNA–mRNA Pair *AC011483.1*-*CCR7* as a Biomarker and Therapeutic Target for Ischemic Cardiomyopathy

**DOI:** 10.3390/ijms231911994

**Published:** 2022-10-09

**Authors:** Qiao Jin, Qian Gong, Xuan Le, Jin He, Lenan Zhuang

**Affiliations:** 1Department of Veterinary Medicine, College of Animal Sciences, Zhejiang University, Hangzhou 310058, China; 2Institute of Genetics and Reproduction, College of Animal Sciences, Zhejiang University, Hangzhou 310058, China

**Keywords:** bioinformatics analysis, ischemic cardiomyopathy, long non-coding RNAs, immune cell infiltration

## Abstract

Ischemic cardiomyopathy (ICM), which increases along with aging, is the leading cause of heart failure. Currently, immune response is believed to be critical in ICM whereas the roles of immune-related lncRNAs remain vague. In this study, we aimed to systematically analyze immune-related lncRNAs in the aging-related disease ICM. Here, we downloaded publicly available RNA-seq data from ischemic cardiomyopathy patients and non-failing controls (GSE116250). Weighted gene co-expression network analysis (WGCNA) was performed to identify key ICM-related modules. The immune-related lncRNAs of key modules were screened by co-expression analysis of immune-related mRNAs. Then, a competing endogenous RNA (ceRNA) network, including 5 lncRNAs and 13 mRNAs, was constructed using lncRNA–mRNA pairs which share regulatory miRNAs and have significant correlation. Among the lncRNA–mRNA pairs, one pair (*AC011483.1-CCR7*) was verified in another publicly available ICM dataset (GSE46224) and ischemic cell model. Further, the immune cell infiltration analysis of the GSE116250 dataset revealed that the proportions of monocytes and CD8^+^ T cells were negatively correlated with the expression of *AC011483.1-CCR7*, while plasma cells were positively correlated, indicating that *AC011483.1-CCR7* may participate in the occurrence and development of ICM through immune cell infiltration. Together, our findings revealed that lncRNA–mRNA pair *AC011483.1-CCR7* may be a novel biomarker and therapeutic target for ICM.

## 1. Introduction

Ischemic heart disease is one of the most common causes of morbidity and mortality worldwide [1]. Although the prevention strategy is relatively advanced, cardiovascular disease is still the main cause of morbidity and death in the aged. More than 80% of patients with ischemic heart disease are elderly [2]. Those 65 years or older account for over 80% of all acute myocardial infarction-related death [3,4,5]. The lack of bloodstream to the energy-dependent cardiomyocytes leads to ischemic cardiomyopathy, with the severe interruption of perfusion leading to heart failure [6]. The current treatment approaches include interventional surgery and drug therapy, which suppress the symptom without treating the condition. Thus, it is urgent to establish new therapeutic strategies. Atherogenesis always leads to thrombotic occlusion, resulting in the recruitment of immune cells to the vessel wall and myocardial infarction (MI) [7]. It is critical for cardiac repair to have immune cell infiltration and intense germ-free inflammation in response to cardiomyocyte necrotic death after MI. Ischemia and necrotic cardiomyocytes trigger the infiltration of leucocytes, thus initiating the regenerative process to clean irreparable cells and to repair the infarcted area by forming scar tissue to maintain cardiac integrity [8,9,10].

Long non-coding RNAs (lncRNAs) are characterized as transcripts which do not encode proteins and are longer than 200 nucleotides [11], but are still capable of participating in many fundamental biological processes like regulating gene expression levels. Therefore, lncRNAs have critical roles in various pathophysiological events. New evidence indicates that lncRNAs play an important role in the development and progression of cardiovascular diseases [12,13,14], and several studies have reported that the dysregulation of lncRNAs is associated with ICM. For example, Ning Liu et al. found that lncRNA *LncHrt* is dramatically downregulated in infarcted hearts, interacting with *SIRT2* and thereby activating LKB1-AMPK downstream cascades signaling to protect the heart from harmful remodeling responses [15]. Simona Greco et al. identified 14 lncRNAs that are dysregulated in ICM patients, some of which were deregulated in the peripheral blood, indicating their potential as disease biomarkers [16]. So far, although some lncRNAs have been reported to be related to ICM, the mechanism of immune-related lncRNAs specifically regulating ICM initiation and progression remain vague.

The rapidly growing field of bioinformatics analyses has become a powerful tool for predicting disease-associated genes, disease subtypes, and therapeutic targets [17]. An emerging understanding of the potential molecular mechanisms of heart failure was provided by identifying key genes and pathways that might be involved in the progression of heart failure through bioinformatics investigation [18]. One upstream key regulator of ICM, *NFIC*, and its downstream genes (*HOPX* and *TSPAN1*), were identified via bioinformatics and experimental analyses [19].

In our study, we obtained mRNA and lncRNA expression profiles of ICM patients and non-failing controls by downloading publicly available RNA-seq data. Using WGCNA, we constructed a correlation network with the training dataset and identified two ICM-related modules. Furthermore, immune-related lncRNAs within the modules were identified and a competing endogenous RNA (ceRNA) network, which transcripts compete for shared microRNAs (miRNAs) to cross-regulate each other, was constructed with 5 immune-related lncRNAs and 13 immune-related mRNAs. Finally, one lncRNA–mRNA pair (*AC011483.1*-*CCR7*) was verified via validating dataset and cell model, and our results showed that *AC011483.1* may regulate *CCR7* through miRNA and affect immune cells such as monocytes, plasma cells, and CD8^+^ T cells, and finally participate in the immune process of ICM.

## 2. Results

### 2.1. Data Preprocessing and DEGs Screening

GSE116250 contains 13 ICM and 14 NF samples. In total, 1709 differentially expressed genes (DEGs), including 922 upregulated and 787 downregulated genes, were identified using the cutoff (|Log_2_ fold change| > 1 and adjusted *p* value < 0.05) (Appendix A). A DEGs heatmap and a volcano plot are shown in Figure 1A,B.

### 2.2. Construction of Weighted Co-Expression Network and Identification of ICM-Related Key Modules

Weighted gene co-expression network analysis (WGCNA) is an efficient method to characterize gene expression patterns between samples and find gene sets highly associated with clinical diseases according to the expression pattern of genes. In order to find the gene sets associated with ICM, we constructed a correlation network. In our study, a scale-free network was constructed using β = 6 (scale-free R^2^, 0.85) as the soft threshold and we analyzed the network topology with 1–20 threshold weights (Figure 2A). A total of 18 modules were identified by hierarchical clustering and dynamic tree cutting functions (Figure 2B). To identify the clinically relevant modules, we correlated each module MEs with disease traits (ICM and NF). The correlation between the modules and ICM was shown in a heatmap (Figure 2C), which showed that the turquoise module is most negative associated with ICM while the blue module is most positive associated with ICM. In addition, the calculation of module membership versus gene significance further verified the turquoise and blue modules as the key modules associated with ICM (Figure 2D). Finally, the expression of genes of the turquoise and blue modules was shown in heatmaps (Figure 2E,F).

### 2.3. Identification of Immune-Related LncRNAs in the Key Modules

Considering that immunological factors play critical roles in the occurrence and development of ICM, we further investigated the critical factors in the key modules. The list of immune-related genes was downloaded from the Immunology Database and Analysis Portal (ImmPort). We screened immune-related mRNAs by taking the intersection of the immune-related genes list and the genes from the combination of turquoise and blue modules. There were 118 immune-related mRNAs (Appendix A). We carried out co-expression analysis between the expression profiles of the immune-related mRNAs and the expression profiles of the lncRNAs of the combination of turquoise and blue modules. Finally, we obtained 137 candidate immune-related lncRNAs (|correlation coefficient| > 0.8) (Appendix A). The top 50 lncRNA–mRNA pairs ordered by |correlation coefficient| are presented in Table 1.

### 2.4. The ceRNA Network Construction

Based on these results, we assumed that these immune-related lncRNAs act as competing endogenous RNAs (ceRNAs) to participate in ICM progression. ceRNAs are transcripts that cross-regulate each other by competing for shared microRNAs (miRNAs) which can guide Argonaute proteins to the target transcripts to prompt their degradation or suppress their translation by base paring [20,21,22], indicating that the expression levels of lncRNA and mRNA are positively correlated in ceRNAs. The ceRNA network links the function of non-coding RNAs with that of protein-coding mRNAs, thus conducing to coordinate a number of biologic processes and it would contribute to disease pathogenesis if the ceRNA network was perturbed. To test this hypothesis, we constructed a ceRNA network from the turquoise and blue modules according to lncRNA–mRNA correlation levels and the regulation of predicted and experimentally validated miRNA–mRNA/lncRNA. The miRcode database [23] was used to collect predicted and experimentally validated miRNA–lncRNA-interaction information, as well as miRNA–mRNA-interaction information. The following criteria were used to screen lncRNA–mRNA pairs for constructing ceRNA networks: (1) the lncRNA and mRNA must share at least one miRNA; and (2) the lncRNA and mRNA must have positively correlated expression (Pearson’s correlation > 0.8). This resulted in 5 lncRNAs and 13 mRNAs being included in the network (Table 2) (Figure 3A). The interaction of lncRNA–miRNA–mRNA in the network was listed in Appendix A. Moreover, all the lncRNAs and mRNAs except *PDK1* in the network had differential expression levels in NF and ICM samples (*p* value < 0.05) (Figure 3B). 

### 2.5. Validation of the Immune-Related lncRNA–mRNA Pairs 

To confirm the accuracy of the obtained results, we used another ICM RNA-seq dataset, GSE46224, as a validating dataset. Firstly, the expression levels of the five lncRNAs in the network were investigated in the GSE46224, revealing that *LINC00452* was significantly downregulated in ICM patients while *AC011483.1* was significantly upregulated in ICM patients (Figure 4A). Secondly, the validation of the 13 mRNAs in the network showed that *IL17RB* and *ITK* were significantly downregulated in ICM patients while *CCR7* and *MICB* were significantly upregulated in ICM patients (Figure 4B). And all these two lncRNAs and four mRNAs were consistent in the training dataset GSE116250 and validating dataset GSE46224. Among these lncRNAs and mRNAs, there were three lncRNA–mRNA pairs included in the ceRNA network (*AC011483.1-CCR7, AC011483.1-MICB, LINC00452-IL17RB*), and the correlations of two lncRNA–mRNA pairs (*AC011483.1-CCR7, AC011483.1-MICB*) were validated (*p* value < 0.05) (Figure 4C). The dysregulation of the three lncRNA–mRNA pairs (*AC011483.1-CCR7, AC011483.1-MICB, LINC00452-IL17RB*) was further validated in glucose and serum-free 293T cell model, which was used to simulate ischemia in vitro. qRT-PCR was performed to measure the expression levels of lncRNA–mRNA pairs. Among them, the expression level of one pair, *AC011483.1-CCR7*, was consistent with the results in the training dataset and the validating dataset (Figure 4D).

### 2.6. Immune Cell Infiltration

In order to further analyze the immune cell composition of ICM as well as the relationship between immune cell composition and *AC011483.1-CCR7*, we carried out immune cell infiltration analysis. CIBERSORTx is a powerful tool which can calculate the infiltration proportion of 22 types of immune cells based on the LM22 reference matrix. After CIBERSORTx’s calculation, we found that the composition of immune cells was significantly different between ICM and NF (Figure 5A). Plasma cells (*p* value = 0.0057) and T cells gamma delta (*p* value = 0.003) were significantly higher in the ICM. Otherwise, monocytes (*p* value = 0.032) and T cells CD8 (*p* value = 0.0057) were much lower in the ICM (Figure 5B). Notably, the expression of *AC011483.1* was upregulated in ICM (Figure 3B and Figure 4A) and the results of the analyses strongly suggested that it may affect immune status of ICM. Therefore, we further explored whether *AC011483.1* affects the immune cells infiltration. We calculated the Pearson’s correlation coefficient between the immune cell infiltration matrix of GSE116250 and the expression profile of *AC011483.1* and *CCR7* of GSE116250. The results suggested that T cells CD8 (*p* value = 0.005) and monocytes (*p* value = 0.021) had significantly negative correlation with *AC011483.1* whereas T cells CD4 memory activated (*p* value = 0.037) and plasma cells (*p* value = 0.005) had strong positive correlation with *AC011483.1* (Figure 5C). T cells CD8 (*p* value = 0.0001) and monocytes (*p* value = 0.034) had significant negative correlation with *CCR7* while T cells CD4 naive (*p* value = 0.032) and plasma cells (*p* value = 0.009) had strong positive correlation with *CCR7* (Figure 5D). The strong correlation between the expression of *AC011483.1-CCR7* and the dysregulated immune cells indicated that *AC011483.1-CCR7* may affect the occurrence and development of ICM through immune cell infiltration.

## 3. Discussion

In our study, an integrated bioinformatics method was used to analyze clinical ICM RNA-seq data. As far as we know, this study first identified immune-related lncRNA–mRNA pair *AC011483.1-CCR7* as associated with ICM and revealed its potential participation in ICM partly through ceRNAs network.

So far, several studies have used microarray [24] and RNA-sequencing [25] to analyze the transcription of lncRNAs in the heart tissues of patients with ICM, and differentially expressed lncRNAs were identified as potential biomarkers for further analysis. Identifying DEGs is important for transcriptome analysis as DEGs are often highly associated with disease status. However, such a method of prioritizing potential biomarkers might not consistently identify the abnormal gene-regulatory networks associated with specific diseases. In these instances, WGCNA provides a solution to identify the most relevant genes highly associated with clinical diseases. Here, we used unsupervised clustering to construct gene modules and selected the modules that were most relevant to ICM. Indeed, genes, including *CSF1R* [26] and *NLRX1* [27], which were previously reported to be related with ICM genes, were assigned to key modules. *NLRX1*, NOD-like receptors family member X1, has significantly downregulated in intestinal MI/R injury [27], and ablation of the mitochondrial NOD-like sensor *NLRX1* may aggravate severe myocardial ischemia-reperfusion injury by increasing cardiac glucose metabolism and Akt signal impairment [28]. Interleukin 34 (*IL-34*) is one of the ligands of colony stimulating factor-1 receptor (*CSF1R*). In patients with normal cardiac function [29] and chronic heart failure [30], higher serum IL-34 levels were significantly associated with more severe coronary artery disease (CAD).

We carried out co-expression analysis between the expression profiles of the immune-related mRNAs and the expression profiles of the lncRNAs of the combination of turquoise and blue modules to identify immune-related lncRNAs. Competing endogenous RNAs (ceRNAs) are transcripts that compete for shared miRNAs and thus cross-regulate each other [22], and miRNAs are ∼21–23 nt small RNAs which can guide Argonaute proteins to the target transcripts to suppress their translation or prompt their degradation by base paring [20,21]. Thus, only lncRNA–mRNA pairs with positive correlation were used to construct the ceRNA network. Previous studies have reported that lncRNA can bind with methyltransferase family, localizing methyltransferase to the promoter region of the gene and adding methyl to cytosine C to inhibit gene transcription [31]. It is worth noting that some immune-related lncRNAs have strong negative correlation with mRNAs; these lncRNAs may regulate immune-related mRNAs via methyltransferase to participate in the immune process of ICM, but their underlying mechanisms in ICM need further study.

All five of the candidate lncRNAs (*UBE2Q1-AS1, NBR2, C3orf35, AC011483.1, LINC00452*) in the ceRNA network were discovered to be related with ICM for the first time. *UBE2Q1-AS1* is a lncRNA antisense to the *UBE2Q1* gene, which is located on chromosome 1, and it was previously reported to contribute to gastric cancer development [32]. *NBR2* is the neighbor of *BRCA1* gene 2 [33]. In various human cancers, the expression of *NBR2* is dysregulated and associated with clinical outcomes [34,35,36]. Moreover, one study provided the first evidence that *NBR2* is relevant with myocardial hypertrophy by finding that *NBR2* inhibits endoplasmic reticulum (ER) stress and myocardial hypertrophy by modulating the LKB1/AMPK/Sirt1 pathway [37]. *C3orf35*, also known as *APRG1*, was associated with poor prognosis of osteosarcoma and may affect tumor metastasis and immune cell infiltration in osteosarcoma patients [38]. *AC011483.1* is a lncRNA antisense to *KLK6*, *KLK7*, *KLK8*, and *KLK9* genes. One study reported that *AC011483.1* was identified as associated with overall survival of lung adenocarcinoma (LUAD) patients [39]. *LINC00452* was found to be upregulated in ovarian cancer cells as well as tumor tissues, and *LINC00452* could enhance the carcinogenic characteristics such as cell proliferation, migration, and invasion in vitro, and the growth of xenograft tumor in vivo [40]. In the current study, we found that the expression level of the five candidate lncRNAs in the ceRNA network were significantly dysregulated in ICM patients. Further, our co-expression analysis showed that these lncRNAs had strong correlation with immune-related genes such as *IL17RB*, *CCR7*, *MICB*, and *HSPA2*. We speculated that these five candidate lncRNAs affect immune-related genes through miRNA, then affect immune cells, and finally affect the occurrence and process of ICM.

Many immune-related lncRNA–mRNA pairs were found through WGCNA and co-expression analyses, but only one pair (*AC011483.1-CCR7*) was verified by another dataset and experiment, which showed the accuracy as well as importance of this pair. CC-chemokine receptor 7 (*CCR7*) has two chemokine ligands (CC-chemokine ligands 19 (*CCL19*) and 21 (*CCL21*)). Previous studies have reported that CCR7 and its ligands have a critical role in lymphocyte and DC homing to the lymph nodes and intestinal Peyer’s patches [41,42,43]. It is meaningful to be the first study identifying the novel immune-related lncRNA–mRNA pair *AC011483.1-CCR7* (Figure 6) associated with ICM. However, our study only researched the correlation between the lncRNA and mRNA as well as the lncRNA–mRNA pair and ICM; the specific mechanisms of intervention require further study.

The results of immune infiltration analysis showed that the proportion of several immune cells were dysregulated in ICM. The strong correlation between the expression of the *AC011483.1-CCR7* pair and the dysregulated immune cells indicated that the *AC011483.1-CCR7* pair may affect the occurrence and development of ICM through immune cell infiltration. 

There are some limitations in our study: (1) the accuracy of our results and prediction can be improved if there are more samples and more complete clinical information; (2) the sequence of *AC011483.1* is known only in human species. Considering the conservation of sequences, we used 293T cell model to verify our results. However, 293T cells are not cardiomyocytes; human cardiomyocytes as well as myocardium clinical samples of ischemic cardiomyopathy are in need of further study to verify the regulation between the immune-related lncRNA–mRNA pair and ICM.

## 4. Materials and Methods

### 4.1. Data Selection

The RNA-seq datasets GSE116250 [44] and GSE46224 [45] were downloaded from Gene Expression Omnibus (GEO) (http://www.ncbi.nlm.nih.gov/geo/ (accessed on 26 October 2021)). GSE116250 includes heart tissues from non-failing (NF) controls (*n* = 14) and ICM patients (*n* = 13); GSE46224 also includes heart tissues from non-failing controls (*n* = 8) and ICM patients (*n* = 8). We downloaded the expression data of 21 ICM heart tissues and 22 non-failing controls from GSE116250 and GSE46224 expression profiling for further analysis. Detailed information about the datasets can be found in Appendix A. The workflow for the bioinformatics analysis in our study is illustrated below (Figure 7).

### 4.2. Data Preprocessing and DEGs Screening

HISAT2 was used to align reads to the reference genome (GRCh37) [46]. Samtools was applied to convert sequence alignment/map (.SAM) format files into binary alignments/maps (.BAM) format [47], and featureCounts was used for quantifying gene expression [48]. Only uniquely mapped reads were used for expression quantification. According to the workflow, the GSE116250 dataset served as a training dataset to identify important genes while the GSE46224 dataset was used for validation. DEGs were screened using the “DESeq2” package [49] in R software (version 4.1.2) with the cutoff |Log_2_ fold change| > 1 and adjusted *p* value < 0.05.

### 4.3. Co-Expression Network Analysis

The co-expression network was constructed using the “WGCNA” package (version 1.71) in R software (version 4.1.2) with the top 5000 genes with the highest median absolute deviation [50,51]. We constructed the adjacency matrix with an appropriate soft threshold (β = 6) when R^2^ reached 0.85 after calculating the Pearson correlation coefficient between any two genes in the matrix. Hierarchical clustering and dynamic tree cutting functions were used to detect modules (minimum size = 30). The name of each co-expression module was named as the color. Through calculating the *p* value and Pearson’s correlation coefficient of module eigengenes (MEs) and the disease trait of each module, we identified the key modules which are most relevant to ICM.

### 4.4. Identification of Immune-Related LncRNAs

A list of immune-related genes was downloaded from the gene list resources in the Immunology Database and Analysis Portal (ImmPort) (https://www.immport.org/ (accessed on 1 June 2022)) [52]. Since there is only mRNA in the immune-related gene set, we identified the immune-related mRNA in key modules first. The intersection of this immune-related gene set and mRNAs of key modules was obtained and defined as immune-related mRNAs. The immune-related lncRNAs of key modules were screened by co-expression analysis of immune-related mRNAs with the |Pearson’s correlation coefficient| > 0.8.

### 4.5. Construction and Analysis of an Immune-Related LncRNA-Associated Competing Endogenous RNA Network

Immune-related lncRNAs and immune-related mRNAs were used to construct a competing endogenous RNA (ceRNA) network. The miRcode database [23] was used to collect predicted and experimentally validated microRNA (miRNA)–mRNA-interaction data, as well as miRNA–lncRNA-interaction data. Competing lncRNA–mRNA pairs were identified using the following criteria: (1) the lncRNA and mRNA must share at least one regulatory miRNA; and (2) expression of lncRNA and mRNA must be positively correlated (Pearson’s correlation > 0.8). These identified lncRNA–mRNA pairs were used to construct the ceRNA network and visualized using Cytoscape 3.9.1 software.

### 4.6. Construction of Ischemic Cell Model and Cell Culture

Control normal 293T cells were cultured in Dulbecco’s modified Eagle’s medium (DMEM) (GENOM, Hangzhou, China) supplemented with 10% fetal bovine serum (FBS) (ExCell, Suzhou, China), 100 IU/mL penicillin, and 100 μg/mL streptomycin (GENOM, Hangzhou, China). Glucose and serum-free 293T cells were cultured in glucose-free DMEM (GENOM, Hangzhou, China) with 100 IU/mL penicillin and 100 μg/mL streptomycin (GENOM, Hangzhou, China) without FBS. These cells were cultured at 37 °C in a humidified incubator under 5% CO_2_ for 24 h.

### 4.7. Quantitative Real-Time PCR (qRT-PCR)

Total RNA of the cultured cells was extracted using TRIZol reagent (Tsingke, Beijing, China). The HiScript III RT SuperMix for qPCR Kit (Vazyme, Nanjing, China) was performed to synthesize the cDNA with 1 μg of RNA. Then, each 96-well plate well was mixed with 6.5 μL of cDNA (diluted at 1:50), 7.5 μL of qPCR SYBR Mix (Vazyme, Nanjing, China), and 1 μL of primers (10 mM). The primers used in the study are listed in Appendix A.

### 4.8. CIBERSORTx

In order to determine the differences in the composition of immune cells between ICM and normal non-failing controls, CIBERSORTx (https://cibersortx.stanford.edu/ (accessed on 16 May 2022)) was used to analyze whole genome expression profiles. We used the LM22 signature as reference and 500 permutations to obtain the proportion of each type of immune cell in the samples of two groups. The close relationship between lncRNA/mRNA and proportion of immune cells was evaluated using Pearson’s correlation coefficient, and a value of *p* value < 0.05 was considered statistically significant.

### 4.9. Statistical Analysis

R software was used to evaluate the correlation between the lncRNAs and mRNAs as well as the correlation between the lncRNAs and immune cell infiltration using Pearson’s correlation analysis. T-test was used to analyze the difference of RNA expression levels between the control 293T group and treated 293T group. Kruskal-test was used to analyze the difference in gene expression levels as well as the immune cell composition in RNA-seq between the NF group and ICM group. The statistical significance was set at *p* value < 0.05.

## 5. Conclusions

In our study, mRNA and lncRNA expression profiles were investigated in heart tissues from ICM patients through RNA-seq datasets. We used the training dataset to construct a correlation network and identified two ICM-related modules using WGCNA. Furthermore, we identified immune-related lncRNAs within the modules and constructed a ceRNA network with 14 immune-related lncRNA–mRNA pairs including 5 lncRNAs and 13 mRNAs. Finally, one lncRNA–mRNA pair (*AC011483.1*-*CCR7*) was verified via the validating dataset and cell model. Results of immune infiltration analysis showed that *AC011483.1-CCR7* may participate in the occurrence and development of ICM through immune cell infiltration. Based on the findings of this study, the lncRNA–mRNA pair *AC011483.1-CCR7* may be a novel biomarker and therapeutic target for ICM.

## Figures and Tables

**Figure 1 ijms-23-11994-f001:**
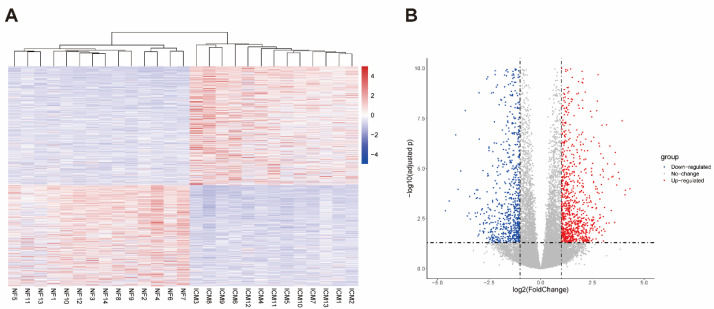
Heatmap and volcano plot to visualize differentially expressed genes identified in the GSE116250 dataset. (**A**) Heatmap of all DEGs. (**B**) Volcano plot. ICM, ischemic cardiomyopathy; NF, non-failing heart; DEGs, differentially expressed genes.

**Figure 2 ijms-23-11994-f002:**
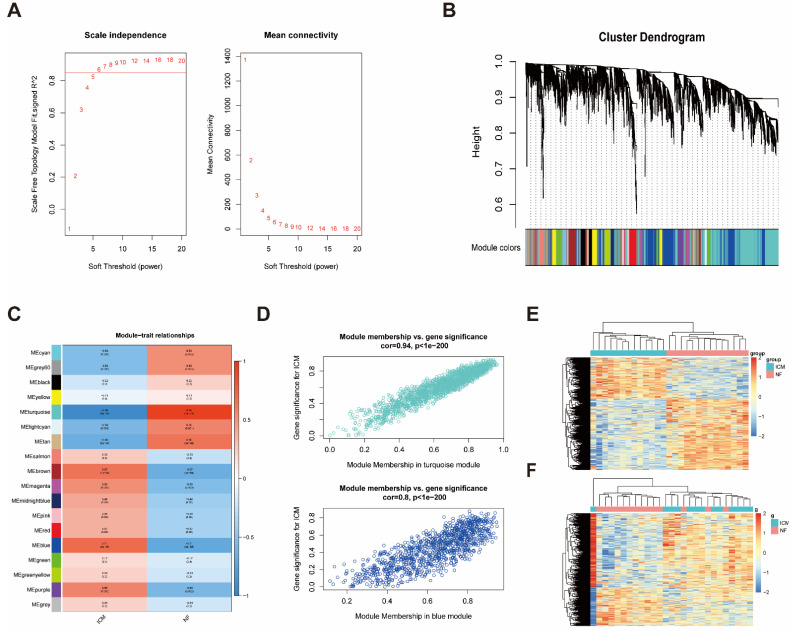
Weighted gene co-expression network analysis (WGCNA) for GSE116250. (**A**) Plot of the scale-free index analysis (left) and mean connectivity analysis (right) for threshold powers of 1–20. β = 6 (scale-free R^2^, 0.85) was chosen as the soft threshold. (**B**) The cluster dendrogram of genes in the WGCNA network, together with assigned module colors. (**C**) Module-clinical trait association analysis. Each column corresponds to a clinical trait, each row to a module. Each grid contains the corresponding correlation (upper) and *p*-value (lower). The table is color-coded by correlation according to the right color legend. The correlation coefficient of module and clinical trait is between −1 and 1, and the association is positive related with the absolute value. (**D**) Scatterplot of GS for ICM versus MM in the turquois (upper) and blue (lower) modules. (**E**) The heatmap of gene expression profile in the turquois module. (**F**) The heatmap of gene expression profile in the blue module. ICM, ischemic cardiomyopathy; NF, non-failing heart.

**Figure 3 ijms-23-11994-f003:**
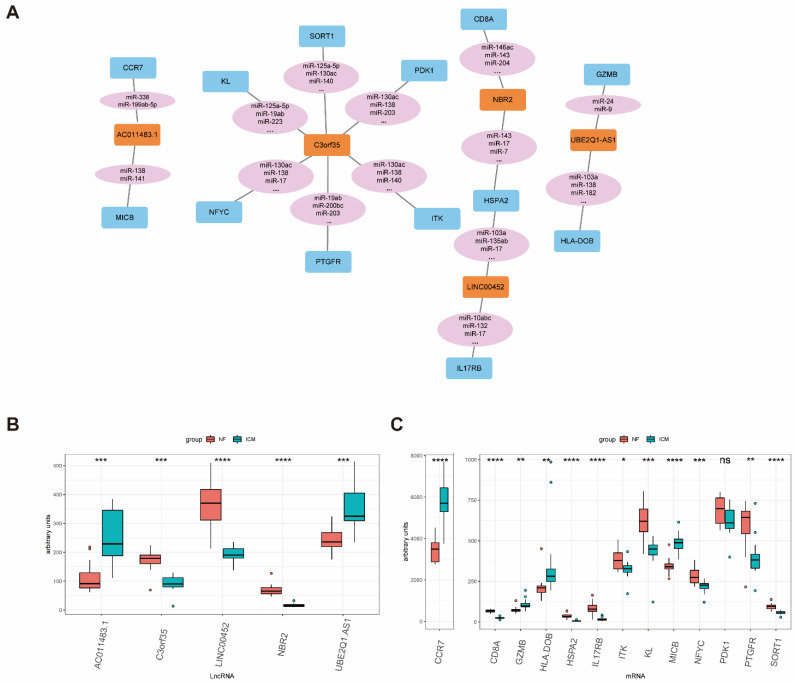
The immune-related ceRNA network within the key modules. (**A**) The immune-related ceRNA network. The blue nodes represent mRNAs, the pink nodes represent miRNAs, and the orange nodes represent lncRNAs. (**B**) Expression of lncRNAs included in ceRNA network among the ICM and NF samples in the training GSE116250 dataset. (**C**) Expression of mRNAs included in ceRNA network among the ICM and NF samples in the training GSE116250 dataset. ICM, ischemic cardiomyopathy; NF, non-failing heart. * *p* value < 0.05, ** *p* value < 0.01, *** *p* value < 0.001, **** *p* value < 0.0001.

**Figure 4 ijms-23-11994-f004:**
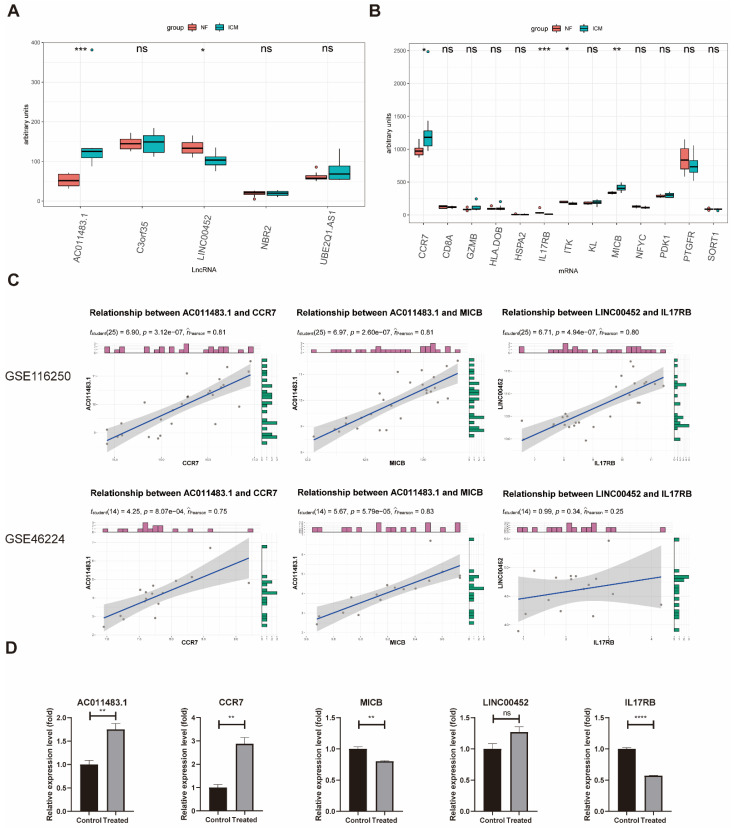
Validation of the expression and correlation of immune-related lncRNA–mRNA pairs. (**A**) Expression of lncRNAs included in ceRNA network among the ICM and NF samples in the validating GSE46224 dataset. (**B**) Expression of mRNAs included in ceRNA network among the ICM and NF samples in the validating GSE46224 dataset. (**C**) Validation of correlations of immune-related lncRNA–mRNA pairs in the validating GSE46224 dataset. (**D**) Expression of lncRNA–mRNA pairs in the glucose and serum-free 293T cell model. ICM, ischemic cardiomyopathy; NF, non-failing heart; Control, normal cultured 239T cells; Treated, 239T cells cultured without glucose and serum. * *p* value < 0.05, ** *p* value < 0.01, *** *p* value < 0.001, **** *p* value < 0.0001.

**Figure 5 ijms-23-11994-f005:**
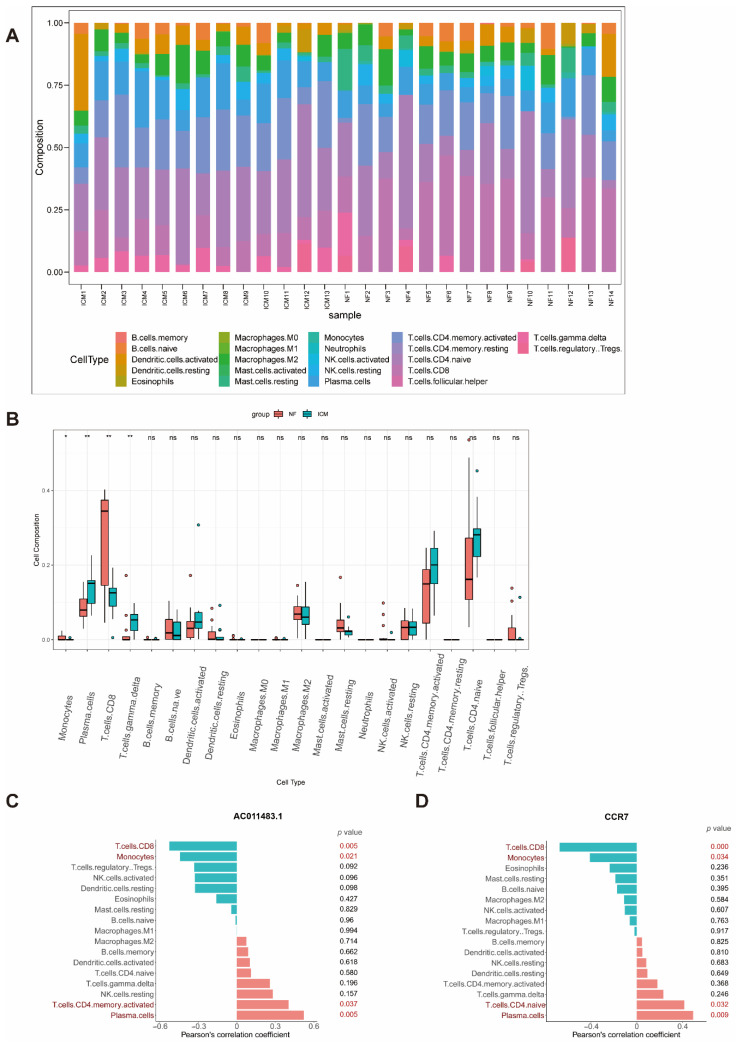
Immune infiltration analysis. (**A**) The proportions of 22 immune cell in ICM and NF samples. (**B**) Differential expression analysis of immune cells in NF and ICM groups. (**C**) Pearson’s correlation analysis of *AC011483.1* and the infiltrating immune cells; statistically significant immune cells were marked in red color with *p* value < 0.05. (**D**) Pearson’s correlation analysis of *CCR7* and the infiltrating immune cells; statistically significant immune cells were marked in red color with *p* value < 0.05. ICM, ischemic cardiomyopathy; NF, non-failing heart. * *p* value < 0.05, ** *p* value < 0.01.

**Figure 6 ijms-23-11994-f006:**
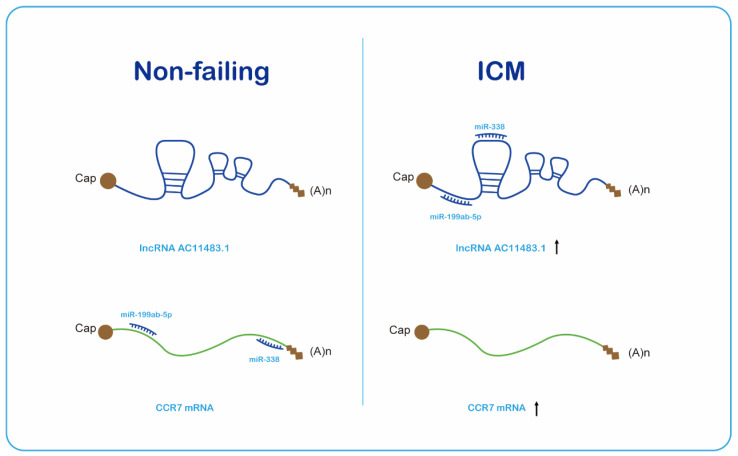
LncRNA AC011483.1 regulates mRNA CCR7 via miRNAs miR-199ab-5p and miR-338.

**Figure 7 ijms-23-11994-f007:**
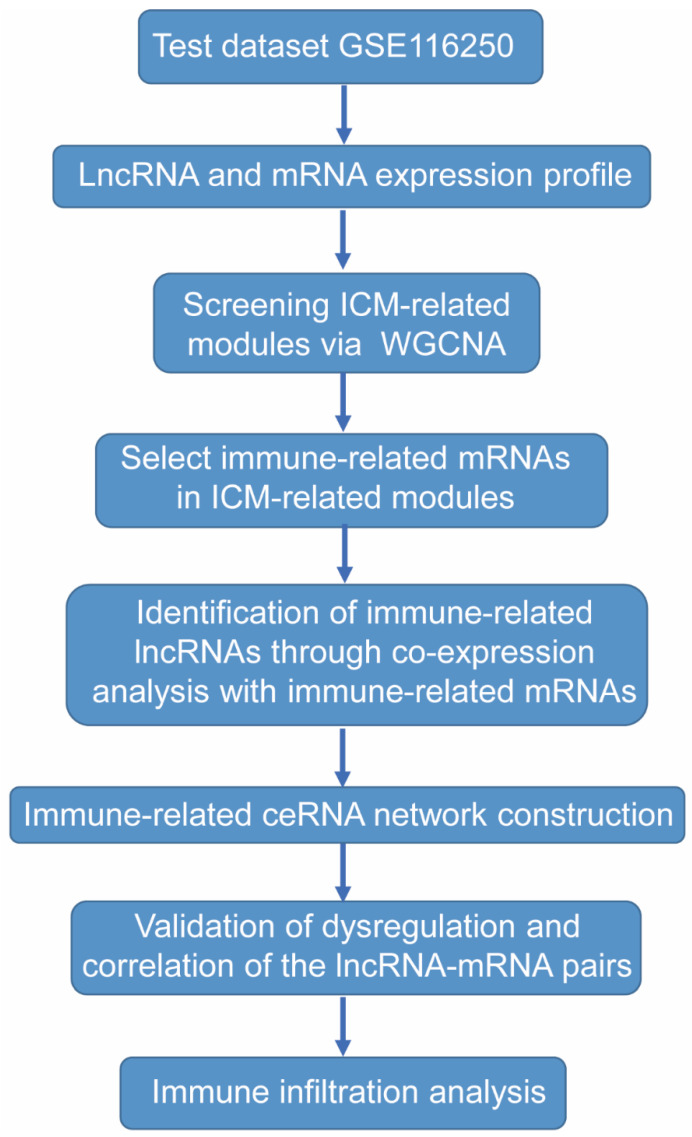
Study flowchart.

**Table 1 ijms-23-11994-t001:** Top 50 lncRNA–mRNA pairs in the co-expression analysis.

Immune-Related mRNA	Immune-Related lncRNA	Correlation Coefficient
*KL*	*C3orf35*	0.938991
*HSPA2*	*CCDC144NL-AS1*	−0.92054
*IL17RB*	*FP700111.1*	0.908619
*IL17RB*	*AL137230.2*	0.904658
*HSPA2*	*AL137230.2*	0.900697
*HSPA2*	*NBR2*	0.898993
*OAS1*	*AL355483.1*	0.893618
*IL17RB*	*AL353576.1*	0.891683
*CD8A*	*AL109924.2*	−0.89132
*SSTR2*	*NUTM2B-AS1*	0.891205
*HSPA2*	*AC018462.1*	−0.88867
*HSPA2*	*FP700111.1*	0.888298
*SSTR2*	*AL357033.1*	−0.88791
*PPBP*	*AL355483.1*	0.886973
*PPBP*	*MYCL-AS1*	0.886181
*JAG2*	*AL445253.1*	0.886144
*HSPA2*	*AC006115.2*	−0.88587
*WFIKKN1*	*AC099811.3*	0.885222
*HSPA2*	*AL139011.1*	−0.88446
*HSPA2*	*AC005776.2*	−0.884
*SORT1*	*AC135050.6*	−0.88333
*IL17RB*	*LINC00958*	0.88294
*HSPA2*	*HCG25*	−0.87913
*CD8A*	*NBR2*	0.879123
*HSPA2*	*AL357033.1*	−0.87882
*CD8A*	*AL109924.4*	−0.87795
*HSPA2*	*LIPE-AS1*	−0.87617
*HSPA2*	*AC100802.1*	−0.87556
*IL24*	*NBR2*	−0.87349
*HSPA2*	*AP002336.2*	−0.87336
*SORT1*	*LINC01788*	0.870983
*CD8A*	*AL136419.2*	0.869864
*HSPA2*	*STAM-AS1*	0.869411
*HSPA2*	*GLYCTK-AS1*	−0.86937
*CD8A*	*AL139011.1*	−0.86919
*SSTR2*	*AC018462.1*	−0.86902
*HSPA2*	*Z84492.1*	−0.86834
*HSPA2*	*AC016727.1*	0.86786
*HSPA2*	*AC008124.1*	0.867474
*IL24*	*AL590666.2*	0.867407
*A2M*	*AC117945.1*	−0.86682
*CD8A*	*CCDC144NL-AS1*	−0.86612
*PPBP*	*KIF9-AS1*	0.865588
*PPBP*	*AC015687.1*	0.865373
*SORT1*	*AL513285.1*	0.864497
*PPBP*	*AC023282.1*	0.864312
*HSPA2*	*AL445253.1*	−0.86361
*HSPA2*	*SEMA3F-AS1*	−0.86289
*A2M*	*AP003721.2*	−0.8627
*HSPA2*	*AC021092.1*	−0.86154

**Table 2 ijms-23-11994-t002:** The lncRNA–mRNA pairs included in the ceRNA network.

Immune-Related mRNA	Immune-Related lncRNA	Correlation Coefficient
*HSPA2*	*NBR2*	0.898993
*CD8A*	*NBR2*	0.879123
*NFYC*	*C3orf35*	0.853143
*ITK*	*C3orf35*	0.828428
*PTGFR*	*C3orf35*	0.801013
*KL*	*C3orf35*	0.938991
*SORT1*	*C3orf35*	0.822644
*PDK1*	*C3orf35*	0.800123
*IL17RB*	*LINC00452*	0.801915
*HSPA2*	*LINC00452*	0.804171
*GZMB*	*UBE2Q1-AS1*	0.819774
*HLA-DOB*	*UBE2Q1-AS1*	0.807795
*CCR7*	*AC011483.1*	0.809768
*MICB*	*AC011483.1*	0.812744
*HSPA2*	*AC011407.1*	0.818169

## Data Availability

Publicly available datasets were analyzed in this study. This data can be found here: [https://www.ncbi.nlm.nih.gov/geo/query/acc.cgi?acc=GSE116250 (accessed on 26 October 2021)] and [https://www.ncbi.nlm.nih.gov/geo/query/acc.cgi?acc=GSE46224 (accessed on 26 October 2021)].

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
