# Peer review of "Bioinformatics and Experimental Analyses Reveal Immune-Related LncRNA–mRNA Pair AC011483.1-CCR7 as a Biomarker and Therapeutic Target for Ischemic Cardiomyopathy"

_ijms, 2022, doi:10.3390/ijms231911994_

Round 1
Reviewer 1 Report
The paper concerns an analysis of expression profiles of lncRNA and mRNA in heart tissues from patients suffering from ischemic cardiomiopathy. The Authors used weighted gene co-expression network analysis and identified a pair of lncRNA-mRNA which can be used as a new marker and/or a potential therapeutic target for ICM.
The paper is interesting and well written. The obtained results are interesting from theoretical point of view and they can also find an application in clinical practice (at least in principle).
Although the Authors used rather standard methods for the performed analyzes, readers not familiar with methods of systems biology and ML can have problems with understanding what exactly was done. So, it would be good if the they shortly explained the idea of the network reated methods they used.
Some minor language errors/typos should also be corrected.
In my opinion after this minor revision the paper could be considered for a possible publication in International Journal of Molecular Sciences.
Author Response
RE: Thanks for your appreciation and constructive comments! According to your advice, we have added “Weighted gene co-expression network analysis (WGCNA) is an efficient method to characterize gene expression patterns between samples and find gene sets highly associated with clinical diseases according to the expression pattern of genes.” in the results 3.2 of the revised manuscript. We added “The ceRNA network links the function of non-coding RNAs with that of protein-coding mRNAs, thus conducing to coordinate a number of biologic processes and it would contribute to disease pathogenesis if the ceRNA network was perturbed.” in the results 3.4 of the revised manuscript.
Reviewer 2 Report
In the current manuscript, Jin et al use publicly available datasets from ICM patients to find a lncRNA-mRNA pair that is positively associated with ICM. Overall, the paper is well written and figures are presented clearly. Some minor comments are below:
1. In the abstract, please provide a line describing the type of dataset downloaded from GEO. Instead of just saying “we downloaded GSE116250 dataset”, please include: “we downloaded a publicly available RNAseq data from ischemic cardiomyopathy patients and non-failing controls (GSE116250)”. Similarly for the other GEO dataset.
2. Please cite the source paper for GSE116250:
Sweet ME et al. Transcriptome analysis of human heart failure reveals dysregulated cell adhesion in dilated cardiomyopathy and activated immune pathways in ischemic heart failure. BMC Genomics. 2018. doi: 10.1186/s12864-018-5213-9. PMID: 30419824; PMCID: PMC6233272.
3. line 71: please define ceRNA
